# The Use of External Fields (Magnetic, Electric, and Strain) in Molecular Beam Epitaxy—The Method and Application Examples

**DOI:** 10.3390/molecules29133162

**Published:** 2024-07-03

**Authors:** Adam Dziwoki, Bohdana Blyzniuk, Kinga Freindl, Ewa Madej, Ewa Młyńczak, Dorota Wilgocka-Ślęzak, Józef Korecki, Nika Spiridis

**Affiliations:** 1Jerzy Haber Institute of Catalysis and Surface Chemistry, Polish Academy of Sciences, Niezapominajek 8, 30-239 Krakow, Poland; bohdana.blyzniuk@ikifp.edu.pl (B.B.); kinga.freindl@ikifp.edu.pl (K.F.); ewa.madej@ikifp.edu.pl (E.M.); ewa.mlynczak@ikifp.edu.pl (E.M.); dorota.wilgocka-slezak@ikifp.edu.pl (D.W.-Ś.); nika.spiridis@ikifp.edu.pl (N.S.); 2PREVAC sp. z o.o., Raciborska Str. 61, 44-362 Rogów, Poland

**Keywords:** MBE, magnetic field-assisted epitaxial growth, magnetite Fe_3_O_4_(111) films on MgO(111), Fe(001) films on MgO(001), UHV resistance measurements, magneto-optic Kerr effect, strain-induced magnetic anisotropy, spin reorientation transition in Pt/Co/Pt films

## Abstract

Molecular beam epitaxy (MBE) is a powerful tool in modern technologies, including electronic, optoelectronic, spintronic, and sensoric applications. The primary factor determining epitaxial heterostructure properties is the growth mode and the resulting atomic structure and microstructure. In this paper, we present a novel method for growing epitaxial layers and nanostructures with specific and optimized structural and magnetic properties by assisting the MBE process using electromagnetic and mechanical external stimuli: an electric field (EF), a magnetic field (MF), and a strain field (SF). The transmission of the external fields to the sample is realized using a system of specialized sample holders, advanced transfers, and dedicated manipulators. Examples of applications include the influence of MFs on the growth and anisotropy of epitaxial magnetite and iron films, the use of EFs for in situ resistivity measurements, the realization of in situ magneto-optic measurements, and the application of SFs to the structural modification of metal films on mica.

## 1. Introduction

Molecular beam epitaxy (MBE) is a powerful tool in modern technologies, including electronic, optoelectronic, spintronic, and sensoric applications [1]. However, along with its advantages, MBE suffers from some limitations connected to (i) demanding ultra-high-vacuum (UHV) environments, (ii) sophisticated sample preparations, and (iii) restrictions in the application of special conditions during the growth of epitaxial heterostructures and their post-growth processing. The growth mode and the resulting atomic structure and micro-structure depend on the elementary epitaxial growth processes, including adsorption/desorption, surface diffusion, nucleation/coalescence, and, finally, the formation of epitaxial films. With the standard MBE technology, these elementary processes are controlled by substrate temperature, deposition rate, and the partial pressure of the gases in cases of reactive deposition. External agents such as plasma generation or ion beams can also be used. Other factors, including external stimuli, such as electric and magnetic fields, which are becoming increasingly common in materials engineering [2], are practically absent in MBE processes under UHVs and during in situ post-deposition treatment. On the other hand, there are examples of the use of external-field-enhanced deposition of thin films in other physical vapor deposition processes, as described below.

Electric fields (EFs): EFs may enhance the surface diffusion according to electromigration. Atomic mobility is related to diffusivity, the strength of the applied EF, and the ion charge, and therefore EF should have significant effects on transport phenomena, as well as on solid-state reactions. The activation energy for surface self-diffusion is much lower than for bulk diffusion (a fraction of eV compared to several eV), and the resulting surface diffusivity is orders of magnitude greater than that in bulk diffusion. Consequently, surface electromigration strongly influences morphology and growth [3]. In particular, it has been shown that in Cu films a few hundred nanometers thick, the activation energy for surface electromigration was 0.9 eV, which led to a measurable effect of the EF on the surface diffusivity in a moderate temperature range of 255–405 °C [4].

The effect of EFs on the growth of thin films has been shown in several semiconductor systems. For example, the crystallinity of ZnO films grown by EF-gradient-assisted pulsed layer deposition (PLD) was remarkably enhanced [5]. EF-assisted sputtering of Fe_3_O_4_ thin films reduced the density of the anti-phase boundaries (APBs) [6,7], which had to have been influenced by the effect of the EF on the nucleation mode. More interestingly, from the point of view of the present paper, in the heteroepitaxy of GaN on 4H-SiC(0001), the electromigration of both Ga and N surface adatoms was observed, which resulted in a composition gradient associated with different effective positive charges [8]. The mechanism of the EF’s effects on the surface morphology of metal surfaces and films was theoretically discussed by Tomar et al. [9]. In this case, the mass transport is accompanied by a high-density electron current, which results in momentum transfer from the electrons to the metal ions.

In conclusion, because MBE is characterized by a low deposition rate, the transient states of the growth process involve single adatoms diffusing over large distances, which makes them particularly susceptible to EFs. Therefore, EFs should affect the growth mode at relatively low temperatures.

Magnetic field (MFs): The mechanism of an MF’s role in phase formation and its stability, albeit in some cases being very pronounced [10], is less recognized in the deposition of thin films. Several studies report the deterministic influence of an external MF on the composition, crystal structure, and magnetic properties of thin metal and oxide films prepared using different methods. Kim et al. [11] found that a continuous Ni catalyst layer on Si(001) was dewetted by post-deposition annealing and agglomerated into well-separated dots, the size and distribution of which were strongly affected by a moderate MF. The microstructure of BiFeO_3_ epitaxial thin films was modified by applying an MF during PLD: a columnar structure was shown in a film prepared under a high deposition rate for a magnetic field of 0.4 T [12]. Nilsen et al. [13] studied the effect of an MF on the atomic layer deposition (ALD) growth of hematite films and observed distinct growth modifications, which were partially ascribed to the weak ferromagnetism of α-Fe_2_O_3_, but they did not explain the physics behind it.

More obvious are the physical mechanism aspects of the influence of MFs on magnetic properties through direct interaction with a magnetic material. Kolotovska et al. [14] reported that paramagnetic organic molecules tend to align their dipole moments with the substrate standard when a moderate external MF (1.3 T) is applied during deposition, which results in a preferential molecular orientation of the film with respect to the substrate. A much weaker MF (~0.05 T) during sputtering was able to induce strong uniaxial in-plane magnetic anisotropy in ferromagnetic (Fe-Sm) films [15], whereas Kuświk et al. [16] established interlayer exchange bias coupling between Co and NiO layers through magnetron sputtering deposition in an MF of 0.11 T. Applying an inhomogeneous MF during magnetron sputtering modified a range of magnetic properties in NiFe/IrMn double layers [17]. Very strong MFs (of several T) are used with PLD processes, but in this case, the role of an MF is to modify the transport of charged versus neutral particles to the substrate [18]. Also, post-deposition annealing under MFs at high temperatures is a routine procedure for shaping the desired magnetic anisotropy (especially for multicomponent alloy and compound films; one of the most prominent examples can be found in Ref. [19]). On the other hand, to the best of our knowledge, the engineering of magnetic properties using MF-assisted MBE remained unexplored until our recent papers exploited the developments described in the present paper [20,21].

Strain field (SFs): MBE growth occurs under no-equilibrium thermodynamic conditions, and therefore it is determined not only by the relative surface and interface energies but, to a great extent, also by the kinetics of the processes occurring between the flux of the deposited atoms (or molecules) and the substrate/film system. Within this subtle balance of different factors, the epitaxial strain/stress, resulting from the lattice mismatch between the film and the substrate, on the one hand, has a profound effect on the structure of the ultrathin films and, on the other hand, constitutes one of the main contributions to magnetic anisotropy. Additional epitaxial strains may arise due to the difference in the thermal expansion coefficients of the deposited material and substrate; for example, upon cooling down from the deposition temperature, in-plane contraction/expansion may lead to tetragonal distortion and break the lattice symmetry. Strains may be responsible for matching the substrate and film lattices, and a reduction in the lattice stress is a driving force for sometimes particular epitaxial relationships. Epitaxial strains play a crucial role in the stabilization of metastable structural phases. As a result, due to magnetoelastic coupling, strains strongly impact the magnetization reversal process, particularly the remanent magnetization. MBE allows for fine control of strains that depend on the film–substrate combination, deposition parameters, and post-deposition processing. Therefore, strains, controlled by lattice mismatch or temperature, are used as a tool to tailor magnetic properties [22,23].

Epitaxial strains lead to bending stresses exerted on the substrate, and by measuring the curvature of the substrate bending during deposition, the surface stress can be monitored [23]. Inversely, many film properties can be modified by bending, a modification extensively studied in epitaxial films grown on flexible substrates, such as mica [24]. Going a step further, deposition can be performed on a bent substrate, and the induced SF in the deposited film should allow for control over phase [25] and magnetic [24] transitions.

In this contribution, we describe the development of the MBE technology, implemented in a research-oriented UHV system, that allows for the application of these external stimuli, EFs, MFs, and SFs, during the MBE growth and post-growth treatment. We describe a unique sample environment that can be used not only to control the MBE growth according to additional degrees of freedom but also to extend in situ specialized characterization methods using such valuable tools as transport measurements or the magneto-optic Kerr effect (MOKE).

## 2. General Concept of the External Field-Assisted MBE System

MBE requires a UHV environment, which is indispensable for the purity and high structural quality of the grown samples. Typically, a base vacuum of 2 × 10^−10^ mbar or greater that remains in the 10^−10^ mbar range when the vapor sources are activated ensures impurity-free deposition, within a typical growth rate of single angstroms per minute. Such strict conditions pose severe limitations on the materials and processes applicable within MBE, and therefore the use of external stimuli (in particular, electric and magnetic fields combined with an elevated temperature), the critical technology in the planned experiments, is not trivial. The idea of the planned experiments is based on the concept of modular sample holders transferable between stations specialized for a given step of the MBE process: substrate preparation, film deposition, post-deposition treatment, or in situ characterization.

### 2.1. Sample Holders

Sample holders are the essential components of any MBE system. The primary feature of a sample holder is its ability to be transferred between preparation/measurement stations in other chambers or systems. In each such place, there is a manipulator that allows for the positioning of the sample and includes heating and cooling options. The current solution combines valuable features of two broadly used types of sample holders: PTS [26], a versatile functional base, and FLAG [27], a simple substrate plate. Both, shown as basic versions in Figure 1, can be used as autonomous substrate holders that fit into the dedicated station of a manipulator. The PTS holders typically include heating options, resistive or electron bombardment (EB) heaters, and temperature sensors accessible thanks to six electrical contacts with the manipulator station. Additionally, they are prepared for the cooling option using a detachable cooling finger of the manipulator. FLAG holders typically do not have heaters and rely on manipulator heating/cooling devices, but they are small in size (have a low outgassing rate) and have a quick thermal response. They are a popular standard in research (e.g., in STM microscopes) and in synchrotron UHV systems.

Both holder types can be transferred under UHV conditions between different distant UHV locations using vacuum transport “suitcases” [28], which we use, for example, for measurements at the National Synchrotron Radiation Center “Solaris” in Krakow [29].

PTS sample holders are specialized for different functions, as described in the next section. They can be used to directly fix the substrate onto it or as adapters accepting FLAG holders, as shown in Figure 2, for a simple PTS/FLAG combination.

### 2.2. Sample Stations, Manipulators, and the Transfer System

The key idea of the proposed solution is the interchangeability of the PTS and FLAG holders so that each standard can be used on its own. Most importantly, the synergy effect of their combination is a meaningful added value for their functionalities. Implementation of this solution requires manipulator stations dedicated to both holder standards and specialized devices for transferring and combining the holders. This way, samples mounted onto FLAG and PTS holders can be transferred between different manipulators, and a sample on a FLAG holder can be inserted into a specialized PTS holder. For this purpose, different manipulators are used, depending on the required functionality in the given position of our MBE system. Along with single-station (PTS [30] or FLAG [31]) manipulators, a dual-station manipulator was developed, as shown in Figure 3.

The PTS station is equipped with six detachable contacts corresponding to those in the PTS holders (two for a heater and four for a pair of thermocouples: C-type and K-type), the liquid nitrogen cooling finger, and four additional side electrical contacts, which are dedicated to providing an electrical connection directly to samples. The high-temperature range depends on the PTS type and reaches 1300 K for resistive heaters and 2000 K for EB heating.

The FLAG station has EB heating and contains a control thermocouple, which should be calibrated to the actual sample temperature using a pyrometer.

Typically, the manipulator is a four-axis high-precision unit that allows for three perpendicular translations and rotation around the main long translation axis.

The dual sample holder system requires at least two specialized transfer linear manipulators or wobble sticks. However, UHV systems typically have several chambers, and the transfer system must be constructed accordingly. Because the functionality of the PTS/FLAG combination depends on the availability of the holders, it is good practice to keep a number of them under UHV conditions. Figure 4 shows examples of the developed storage for PTS and FLAG holders, which can be mounted into UHV systems using a single CF64 port.

### 2.3. Description of the UHV System

Figure 5 shows a top view of our UHV MBE apparatus, adapted to the dual sample holders by modifying the manipulators and the transfer system. The UHV system contains three main chambers: the MBE chamber, the STM chamber, and the auxiliary chamber for pre-treatment of the PTS holders and conversion electron Mössbauer spectroscopy (CEMS) measurements. Two load-lock entrance chambers, one for each holder type, also allow us to attach vacuum transportable suitcases. The base pressure in the system is below 2 × 10^−10^ mbar.

The MBE chamber includes a home-built deposition system with six metal vapor sources (typically Fe isotopes, Co, Au, Ni, and Pd) using resistively heated BeO crucibles and EB evaporators for Pt and MgO. The vapor sources are embedded into a water-cooled Cu shield, which ensures deposition of the pure elements under a 10^−10^ mbar pressure range at rates of several angstroms per minute. The deposition rate is controlled using a quartz crystal monitor with ±5% accuracy. The oxygen dosing system enables the reactive deposition of metals to grow simple oxide compounds. Using the dual-station manipulator, the MBE chamber accepts both holder types. A four-grid optics system (OCI Vacuum Microengineering Inc., London, ON, Canada) for low-energy electron diffraction (LEED) and Auger electron spectroscopy (AES) is available in this chamber.

The second chamber is dedicated to scanning tunneling microscopy (STM, Burleigh Instruments, Fishers, NY, USA), which was adapted to the FLAG holders.

The third chamber, with the PTS station manipulator, contains a home-built CEMS spectrometer [32], which is a useful characterization tool for iron-containing ultrathin films, especially when the ^57^Fe isotope is used for their preparation.

## 3. Specialized Sample Holders

Magnetic, electric, and mechanical stimuli are applied to the sample using a combination of magnets, coils, electrodes, and actuators that are incorporated into the dedicated PTS holders. A number of PTS holders play the role of adapters for the FLAG holders, and the FLAG sample plate with a substrate can be moved between different PTS adapters for subsequent preparation steps (cleaning, deposition, annealing) performed under variable external stimuli.

Moreover, external fields can be used during in situ electrical and magnetic characterization of the samples: transport measurements and MOKE magnetometry.

### 3.1. Sample Holders with Magnetic Fields

Two types of PTS adapters with magnetic fields were developed, the parameters of which are described below and summarized in Table 1.

#### 3.1.1. Constant Magnetic Field PTS Adapters

The concept of constant MF holders was focused on epitaxial growth and post-deposition annealing using two geometries of MFs: in-plane and out-of-plane. The solution is based on stray fields of Sm-Co permanent magnet arrays in two arrangements, as shown in Figure 6.

Two versions of the solutions were optimized: (i) for the maximum MF and (ii) for the combination of an MF with high temperatures. In the first version (Figure 7a,b, for the out-of-plane and in-plane MFs, respectively), the FLAG plate is placed in direct contact with the magnet, which is assembled from five to ten cylindrical or rectangular Sm-Co segments. The second version (solution details for the out-of-plane MF are shown in the cross-section in Figure 7c) incorporates a Ta resistive heater, thermally isolated from the magnets. To prevent the magnets from degradation (the maximum operation temperature should not exceed 200 °C), they can be cooled using the LN_2_ cooling option in the PTS station. The sample and magnet temperatures are independently controlled using two thermocouples. In this way, sample temperatures up to 600 °C can be reached safely.

One important feature of the solution is the modular construction of the PTS adapters, which ensures their universality and easy maintenance (e.g., the heater or magnet exchange).

#### 3.1.2. PTS Adapters with Small Electromagnets

The PTS adapters with variable MFs produced using a small electromagnet with a soft iron core were inspired by the solutions used in the PEEM microscopes installed in the BESSY II, SLS, and ALBA synchrotrons [33,34,35]. The electromagnets were optimized for a maximum MF that enables MOKE measurements. The optimization uses LN_2_-cooled low-ohmic coils of Kapton^®^ isolated wires, thermally isolated from a sample heater, similar to those described above for the permanent magnet PTS adapters. Photos of the variable MF adapters ensuring field geometries for polar MOKE (P-MOKE), longitudinal MOKE (L-MOKE), and transversal MOKE (T-MOKE) are shown in Figure 8a, Figure 8b, and Figure 8c, respectively.

Table 1 below summarizes the parameters of the MF PTS adapters used for deposition and MOKE measurements. The “Magnetic Field Range” data are for reference only. The exact magnetic field induction and its distribution were measured in the air for a particular adapter and, in the case of the variable field holders, calibrated against the coil current. For the permanent magnet adapters, the MF uniformity contributed the most to the characteristic uncertainties, which for a sample area of 5 × 5 mm^2^ did not exceed 5%. Additionally, the thickness of the inserted FLAG–substrate combination contributed to an additional minor MF uncertainty. The maximum magnetic field, its uniformity, and the uncertainty for the electromagnet solution with an in-plane MF are determined by the electromagnet gap. The values in Table 1 are for a 10 mm gap, ensuring 3% field uniformity, sufficient for measuring wedge samples along a 5 mm distance, as was verified through the comparison of MOKE measurements using the PTS adapter and a laboratory electromagnet. A reasonable reduction in the gap to 5 mm allows the MF induction to be doubled at the expense of uniformity.

Among the specialized PTS adapters, the one with rotation of the inserted FLAG holder around the perpendicular axis using a wobble stick should be mentioned in this context, as it is useful for angular L-MOKE measurements.

### 3.2. Sample Holders for Electrical Measurements

The PTS and FLAG holders can be biased up to 1000 V in the dedicated manipulator station, thanks to which deposition can be performed under an EF, whose characteristics would depend on the geometry of the counter electrode [5]. To apply in-plane EFs or for transport measurements, the additional four contacts with the PTS station of the manipulator in the MBE/LEED chamber are used. They allow for the transmission of electric signals to the sample mounted onto the PTS holder with the counter male contacts made via spring connections, as shown in Figure 9. Different forms of spring connections were tested, depending on the size of the sample substrate, the required temperature range, and the measurement type. The photo in Figure 9 shows the tungsten wire connections to molybdenum stripe pads pre-deposited onto an MgO substrate.

### 3.3. Sample Holder for Substrate Bending

The PTS manipulator station has the function of actuating cooling by pressing the cold finger against the PTS holder. This function was used to bend the substrate, as shown in Figure 10a. The moving piston (marked with an arrow) pushed by the cold finger pressed the flexible substrate, resulting in a change in its curvature. The substrate can be bent during sample preparation and growth processes, as well as after growth. With different substrate mounting configurations, bending with compressive or tensile stress can be achieved. The photo in Figure 10b shows the realization of this with two clamps that facilitate bending with a positive film curvature. The maximum curvature is determined by the piston stroke and the clamp distance, which are 0.5 mm and 6 mm, respectively, which gives a minimum curvature radius as small as approximately 10 mm. Notably, the presented solution preserves all the thermal functionalities of the holder (100–1300 K).

## 4. Application Examples

### 4.1. Epitaxial Films Grown under a Magnetic Field—Fe_3_O_4_(111)/MgO(111) and Fe(001)/MgO(001)

To verify the role of applying an MF during growth, we selected two model systems: archetypic Fe(001) films on MgO(001) [21] and reactively grown Fe_3_O_4_(111) films on MgO(111) [20].

Fe(001) films (at a typical thickness of 10 nm) were deposited at room temperature and annealed at 673 K, and the entire preparation process was performed using an in-plane MF of 100 mT. In situ STM images revealed the distinctly different morphologies of the film grown under the applied field and of the control sample grown with no field, as shown in Figure 11a and Figure 11b, respectively. Ex situ magnetic characterization using Kerr magnetometry and microscopy proved the magnetization reversal via 90° domains for both the no-field and in-field samples, evidencing clear differences in the size and shape of the magnetic domains correlating with the film morphology [21].

The significant impact of an MF on the magnetic properties was also shown for ultrathin epitaxial magnetite films grown on MgO(111) [20]. Using in situ STM and CEMS and ex situ MOKE, we showed that a moderate MF of 0.1 T applied in plane during the reactive deposition of a 10 nm Fe_3_O_4_(111) film induced a sizable perpendicular magnetic anisotropy. This effect is illustrated in Figure 12. The spectrum in Figure 12a corresponds to a reference sample deposited with no magnetic field. The green dashed line marks the intensity relation between the second and third line groups. The intensity ratio of this line group (as well as of the fifth and fourth line groups) reflects the average direction of the spontaneous magnetization with respect to the gamma rays, and for the reference spectrum, it indicates that the magnetization is evenly distributed along the four easy axes in the cubic structure of magnetite (for details, compare with Ref. [20]). For the sample deposited under an in-plane MF (spectrum b in Figure 12), the marked intensity ratio is distinctly changed, which witnesses increased perpendicular magnetization. It is clear that the MF being applied in-plane during deposition enhanced the perpendicular anisotropy. Additionally, the spectrum in Figure 12c measured in an external magnetic field of 0.4 T applied along the film normal exhibited a further decrease in the intensity of the second line group. The numerical analysis indicated that this field was almost enough to saturate the magnetization, which was in agreement with the MOKE measurements [20]. Similar to the Fe(001) films, an MF being applied during deposition had a visible impact on the morphology of the magnetite films.

The examples above illustrate the impact of applying an MF during deposition both on the microstructure and magnetic properties of the resulting films. We attribute the morphology–magnetism interplay revealed in these experiments to magnetostrictive effects.

### 4.2. MOKE—Spin Reorientation Transition in Pt/Co/Pt

MOKE is one of the most used and useful methods for characterizing the magnetic properties of magnetic thin films. It is relatively simple and inexpensive when performed in ambient conditions. On the other hand, in situ MOKE measurements under UHVs require transmission of the magnetic field to the sample, which is challenging due to the size and space occupied by an electromagnet. Therefore, PTS adapters with variable magnetic fields, as described in Section 3.2, are attractive solutions for implementing MOKE measurements under UHVs.

The principle of MOKE measurements for two basic geometries, longitudinal (L-MOKE, magnetization vector parallel to the film plane and parallel to the plane of incidence) and polar (P-MOKE, magnetization vector perpendicular to the film plane and parallel to the plane of incidence), are shown in Figure 13a and Figure 13b, respectively. The method relies on a change in the light’s linear polarization (rotation and ellipticity) reflected from a magnetized film, and the required sensitivity for monolayer films, of a fraction of a milliradian, is achieved using a modulation technique. In our solution, the implementation of the MOKE measurements requires only one (for P-MOKE) or two (for L-MOKE) viewports to be centered on the sample.

The PTS adapters described in Section 3.2 can deliver the magnetic fields required for all MOKE geometries; however, their application is limited to samples with moderate saturation fields not exceeding 100 mT (compare Table 1).

As an example of an application, we present results on MOKE measurements on a Pt/Co/Pt/MgO(111) heterostructure, in which the thickness of the Co film changes gradually from 0 to 2 nm. Such a wedge sample was obtained using a shutter moving in front of the sample during Co deposition. This system is known to exhibit a spin reorientation transition (SRT) from out-of-plane magnetization to in-plane magnetization with a critical thickness of approximately 1 nm [36]. The P-MOKE loops presented in Figure 14 were obtained by precisely moving the laser spot along the sample in the wedge direction of the sample. Remarkably, the sensitivity of the presented solution is sufficient to acquire a MOKE loop for the film as thin as 0.3 nm. The presented loops show that the cobalt films exhibit perpendicular magnetization below 0.6 nm, at which thickness the SRT begins. For thicker films, the accessible magnetic field is not strong enough to saturate the magnetization out of the plane. Although the P-MOKE intensity is the highest, monolayer sensitivity is also achievable for longitudinal and transversal geometries.

### 4.3. Transport Measurement—In Situ Resistivity of Magnetite Ultrathin Films

Magnetite is an exceptional compound due to its combination of magnetic and transportable properties, which is rare for oxides. Its quasi-metallic conductivity at elevated temperatures abruptly drops by two orders of magnitude across the Verwey transition at around 125 K. In parallel, its high Curie temperature makes magnetite thin films interesting for spintronics, but their magnetic and electrical properties undergo significant modifications with a decreasing thickness. It has been reported that APBs, which strongly influence magnetic properties, are also responsible for a significant reduction in electrical conductivity [37]. Our in situ resistivity measurements below suggest an additional origin of this effect.

Our research was motivated by comparing in situ and ex situ CEMS measurements on ultrathin magnetite films grown under UHV conditions. We found that these films oxidize toward maghemite when exposed to the atmosphere. For this reason, to reveal the true character of the thickness dependence of the resistivity in magnetite, the electrical measurements should be performed in situ under a UHV. This was possible using special sample holders, as described in Section 3.2.

Epitaxial magnetite Fe_3_O_4_(001) films were grown on MgO(001) by the reactive deposition of Fe under the O_2_ partial pressure of 5 × 10^−6^ mbar on substrates kept at 530 K. Figure 15 shows the thickness dependence of the resistance measured in situ during growth from a 0 to 10 nm thickness. The resistance starts to decrease at a thickness as low as 0.6 nm and drops by over five orders of magnitude into the several kΩ range. The resistance is sensitive to molecular oxygen adsorption, which we interpret as surface oxidation toward maghemite. This process does not saturate at an in situ exposure of 400 Langmuir at the 10^−6^ mbar O_2_ partial pressure and further proceeds after exposure to the ambient atmosphere, as shown in the inset in Figure 15, which documents a 25% increase in the resistivity during 16 h of air exposure. The increase in the resistivity of this order, measured ex situ, was interpreted as originating from the APBs.

### 4.4. Bending of the Substrate—Induced Uniaxial Anisotropy in Co Films on Gold

To test the influence of the strain caused by bending on the magnetic properties, we selected epitaxial cobalt films on Au(111)/mica. This system is known for cobalt’s self-organization, directed by the herring-bone reconstruction of gold that occurs in three 120-degree domains [38], which should result in three-fold symmetry in the magnetic properties. It is also known that the bending of mica, equivalent to uniaxial external stress, causes restructuring of the herring-bone network: the three-fold orientation degeneracy of the characteristic reconstructing pattern is removed [39]. Following the above findings, we grew a 40 nm Au film on mica. A LEED pattern after annealing the film at 673 K (Figure 16a) indicates the good crystallinity of the gold but without a visible herring-bone reconstruction pattern. Then, using the dedicated PTS holder, as described above, an approximately 0.5% compressive strain was achieved by bending the mica. On such a bent substrate, a 2 nm Co film was deposited, exhibiting a weak LEED pattern that documented the epitaxial relations between Co and Au. Finally, for ex situ measurements, the sample was covered (also under strain) with a 2 nm Au layer.

Figure 16b shows the results of ex situ MOKE measurements of the Au/Co/Au/mica heterostructure after the bending was released. The polar plot represents the angular dependence of the coercivity extracted from the set of longitudinal MOKE curves, which are exemplified by the loop in the center of the polar plot.

Obviously, the coercivity exhibits two-fold symmetry, indicating uniaxial anisotropy induced when the Co film was relaxed after releasing the strain. The symmetry axis of the polar plot in Figure 16b (at an angle of 30° from the bending axis) coincides with one of the three possible easy axes in the Co hexagonal plane.

## 5. Conclusions

The developed sample holders provide numerous new functionalities for an MBE system, as shown in application examples related to electronic and spintronic systems. The external stimuli integrated into the adapters and the sample holders allow us to extend the sample environment during the MBE process and expand the range of in situ experimental tools.

The presented solutions can be easily implemented in existing UHV systems and allow for the ingenious combination of the PTS and FLAG holder standards. The developed research-laboratory-oriented solution has some direct application potential; however, adapting the present technology to industrial applications might be challenging due to the requirements imposed by large-sized substrates.

We expect that the developed methodology will help precisely tailor the functionality of epitaxial nanostructures for a wide range of applications, particularly spintronics. Our research hypothesis is that the activation of these external stimuli during MBE growth will affect many structural, electronic, and magnetic properties. The background of our hypothesis is a high susceptibility of the MBE process to external parameters on the one hand and the strong dependence of the physical parameters on the system size and dimensionality, known as size effects, on the other hand. Thus, when an external stimulus is applied during film growth, it is plausible that the critical temperature (e.g., the Curie or Neel temperature) for an ultrathin film will fall within the deposition temperature range, resulting in the enhanced impact of the applied field. The set of materials in the scope of our interest and under current investigation include iron oxides (hematite, magnetite) and metal oxide heterostructures, including a ferromagnetic layer with perpendicular magnetic anisotropy and with an antiferromagnetic layer responsible for the exchange bias effect. The research topics currently in progress are, for example, tuning the structural phase transition in hematite by MF-assisted growth, controlling the growth of epitaxial oxide films on flexible substrates using EFs and SFs, and tailoring perpendicular magnetic anisotropy in epitaxial heterostructures using MF-assisted MBE. Flexible oxide films (free-standing or deposited on flexible substrates) are an emerging promising technology in electronics. Spinel films (magnetite or cobalt ferrite) on mica are among the most studied epitaxial systems [24]; however, detailed analysis of the role of strain/stress on their growth, magnetoelastic, and transport properties is lacking. Using in situ bending technics, we will be able to quantify these effects and contribute to a fundamental understanding of the mechanism behind them.

## Figures and Tables

**Figure 1 molecules-29-03162-f001:**
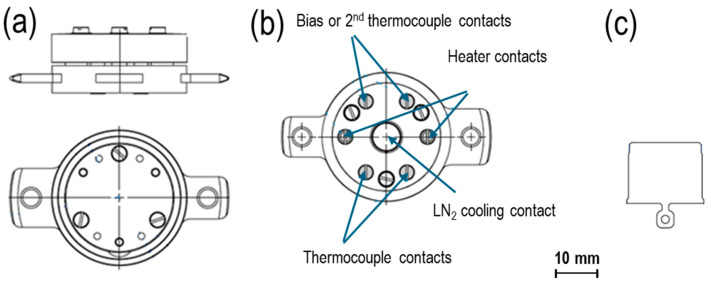
Geometry of the sample holders: (**a**) side and top views, (**b**) bottom view of PTS, (**c**) FLAG holder.

**Figure 2 molecules-29-03162-f002:**
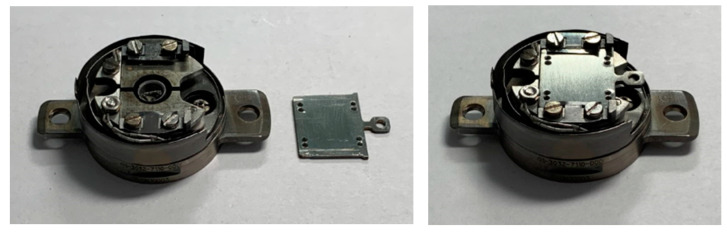
Example of a PTS/FLAG combination: detached (**left**) and assembled (**right**).

**Figure 3 molecules-29-03162-f003:**
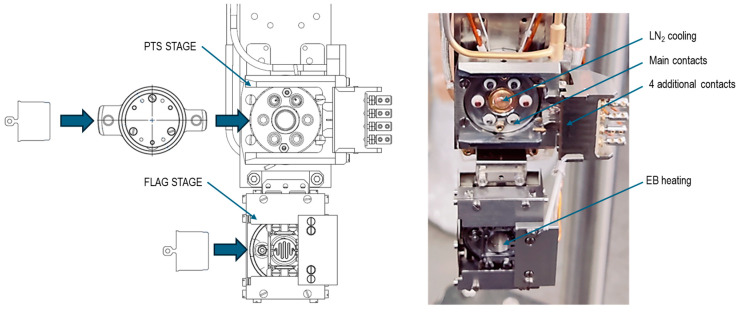
The holder area of the dual-station manipulator with the PTS and FLAG holders is shown (**left**), and a corresponding photo (**right**).

**Figure 4 molecules-29-03162-f004:**
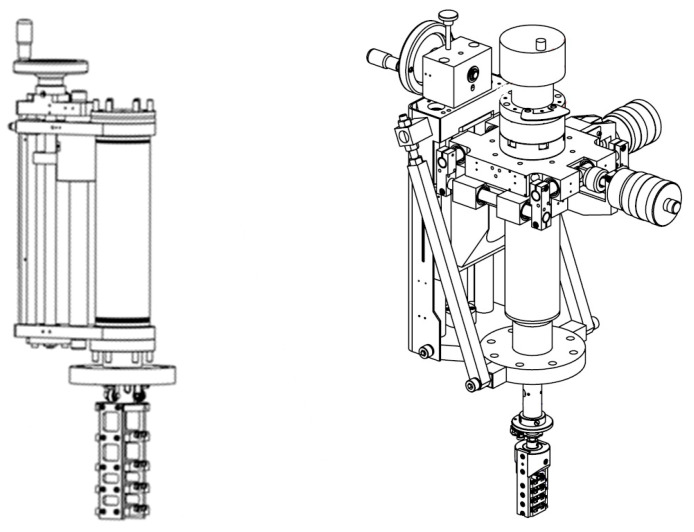
Four-position PTS storage mounted onto a linear manipulator (**left**), and linear four-position FLAG storage mounted onto a four-axis manipulator (**right**).

**Figure 5 molecules-29-03162-f005:**
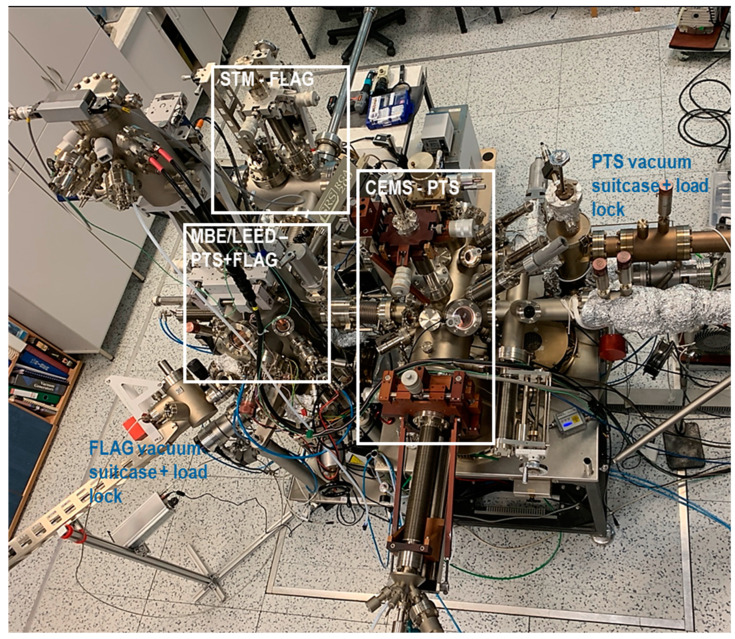
Top view of the UHV system.

**Figure 6 molecules-29-03162-f006:**
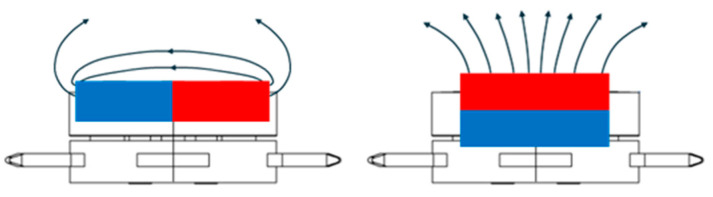
Schematics of the constant MF PTS adapters based on stray fields of Sm-Co permanent magnets for the in-plane field (**left**) and perpendicular field (**right**).

**Figure 7 molecules-29-03162-f007:**
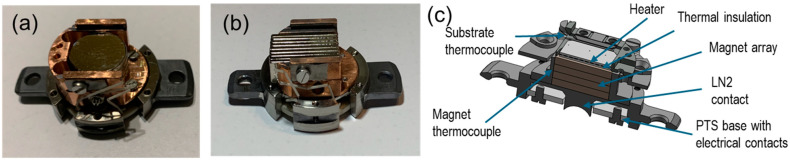
Photos of PTS adapters for applying MFs with different geometries, out-of-plane (**a**) and in-plane (**b**), without heating options. (**c**) The cross-sections of a PTS adapter for out-of-plane MFs with heating options.

**Figure 8 molecules-29-03162-f008:**
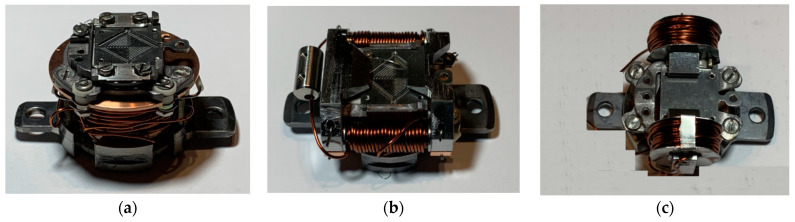
Variable magnetic field PTS adapters for P-MOKE (**a**), L-MOKE (**b**), and T-MOKE (**c**).

**Figure 9 molecules-29-03162-f009:**
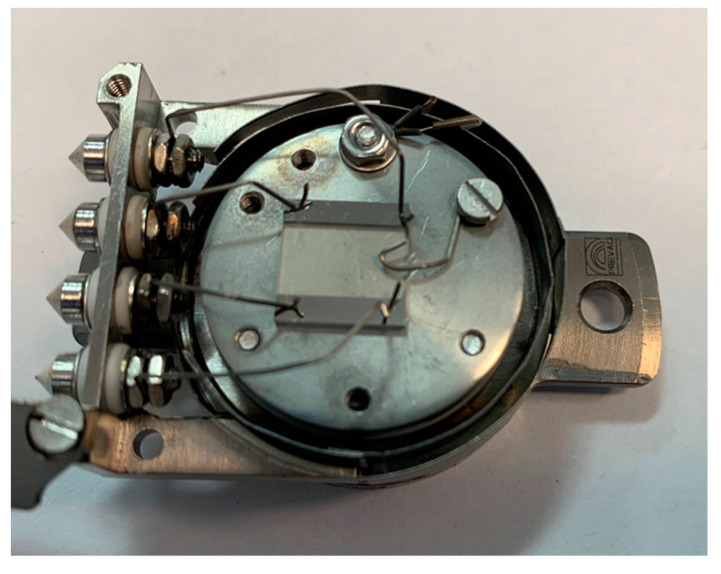
PTS holder with four additional contacts (temperature range of 100–1300 K) and contacting tungsten wire springs to a substrate with pre-deposited molybdenum stripe pads.

**Figure 10 molecules-29-03162-f010:**
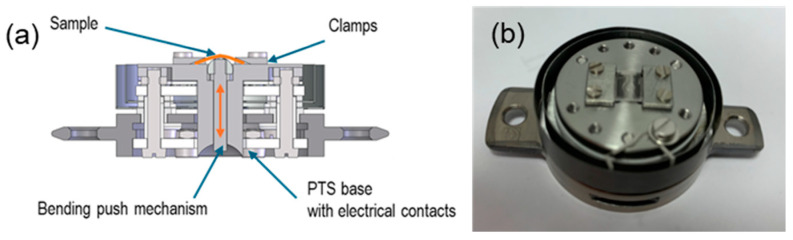
A cross-section of the PTS holder with the bending function (**a**) and its realization (**b**).

**Figure 11 molecules-29-03162-f011:**
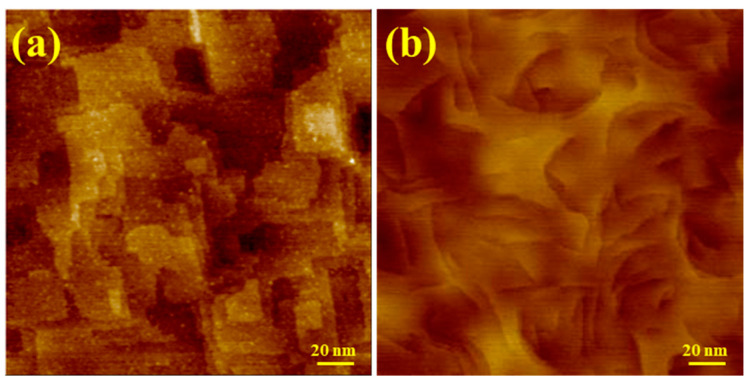
STM images of the Fe(001) on MgO(001) film grown under the applied MF (**a**) and of the control sample grown with no field (**b**). Adapted from [21].

**Figure 12 molecules-29-03162-f012:**
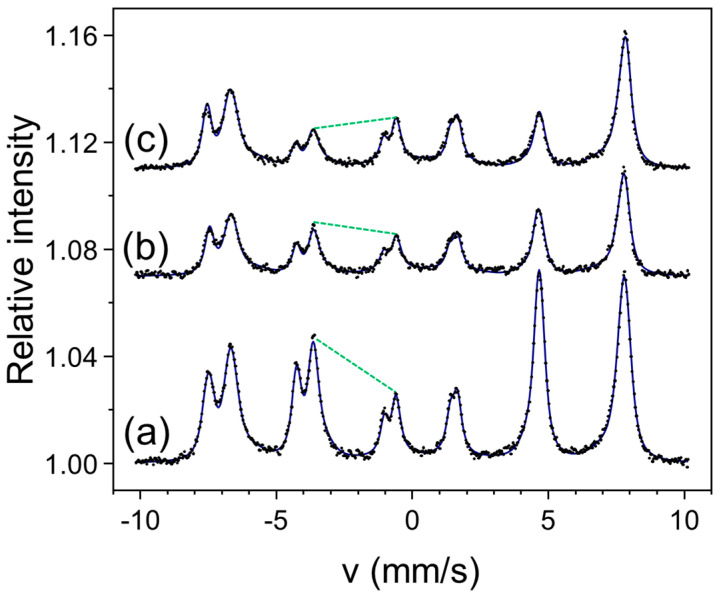
CEMS spectra for 10 nm Fe_3_O_4_(111) films on MgO(111) deposited with no field (**a**) and under an in-plane MF (**b**). Spectrum in (**c**) was measured in an external magnetic field of 0.4 T along the film normal. The dashed green lines mark the intensity relation between the second and third groups of the Mössbauer lines.

**Figure 13 molecules-29-03162-f013:**
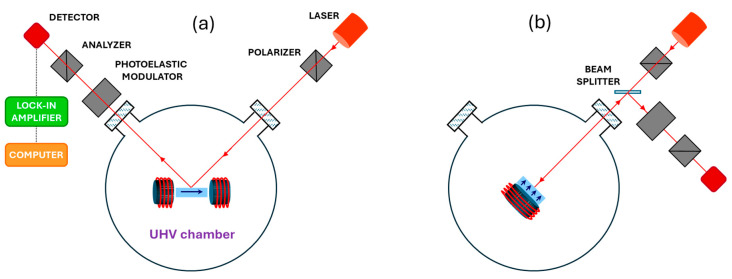
Schematics of the L-MOKE (**a**) and P-MOKE (**b**) setups realized using MF PTS adapters, which are schematically shown in the centers of the chambers. The light beam geometry shown in (**a**) also applies to T-MOKE using the PTS adapter with the MF perpendicular to the incidence plane.

**Figure 14 molecules-29-03162-f014:**
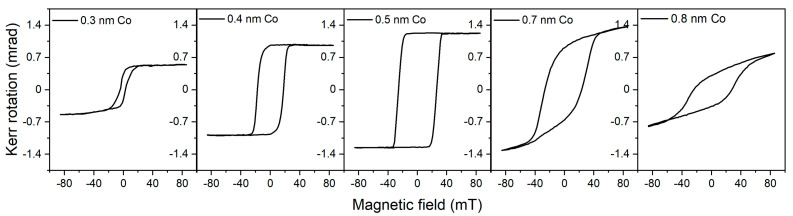
P-MOKE loops for a wedge Co film in a Pt/Co/Pt/MgO(111) epitaxial heterostructure. The loops were measured using a PTS variable magnetic field adapter and the setup from Figure 13b.

**Figure 15 molecules-29-03162-f015:**
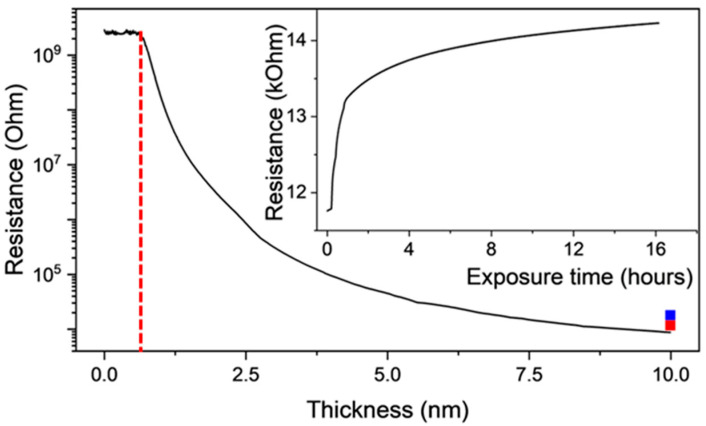
In situ resistance during growth of a Fe_3_O_4_(001) film on MgO(001) with a final thickness of 10 nm. The red dashed line marks a nominal thickness (0.6 nm), at which the resistance change is noticeable. The red and blue points show a resistance increase after in situ O_2_ adsorption (exposure of 400 Langmuir) and exposure to ambient atmosphere, respectively. The inset shows the time evolution of the resistance for the sample exposed to air.

**Figure 16 molecules-29-03162-f016:**
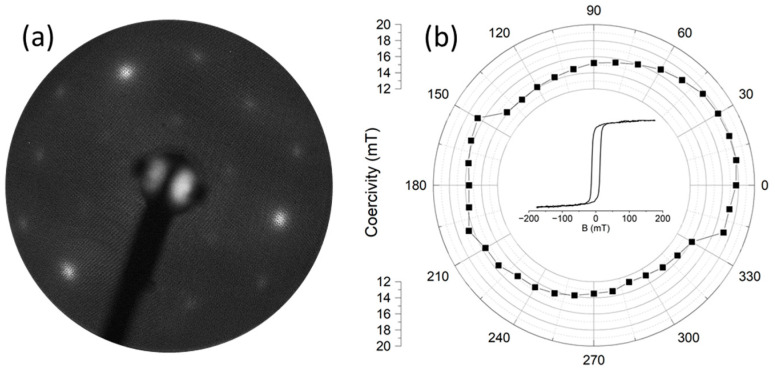
LEED pattern of the 40 nm Au(111) film on mica (**a**) and polar plot of the coercivity extracted from the polar MOKE loops for 2 nm Co film in Au/Co/Au/mica heterostructure (**b**). The inset (**b**) shows an exemplary loop. The errors in the experimental data in (**b**) are less than the point size.

**Table 1 molecules-29-03162-t001:** Parameter overview for PTS adapters with magnetic fields. The parameters of the variable MFs are given for a 10 mm electromagnet gap.

Magnetic Field Source	Magnetic Field Type	Magnetic Field Orientation	Magnetic Field Range	Temperature Range
Sm-Co	constant	in-plane	150 mT	RT
Sm-Co	constant	in-plane	130 mT	120–870 K
Sm-Co	constant	out-of-plane	290 mT	RT
Sm-Co	constant	out-of-plane	140 mT	120–870 K
Coil + Armco	variable	in-plane	±70 mT	120–570 K
Coil + Armco	variable	out-of-plane	±100 mT	120–570 K

## Data Availability

Data are contained within the article.

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
