# Peer review of "The Use of External Fields (Magnetic, Electric, and Strain) in Molecular Beam Epitaxy—The Method and Application Examples"

_molecules, 2024, doi:10.3390/molecules29133162_

Round 1

Reviewer 1 Report

Comments and Suggestions for Authors

The authors' article is written in good language. The material is interesting and may be in demand. There are comments: 1. Authors need to pay attention to the numbering of figures and the way they are presented. 2 Section “Conclusions need to be expanded. 3. The list of cited literature can be shortened without damaging the article.

Author Response

The authors' article is written in good language. The material is interesting and may be in demand. There are comments: 1. Authors need to pay attention to the numbering of figures and the way they are presented. 2 Section “Conclusions need to be expanded. 3. The list of cited literature can be shortened without damaging the article.

We followed the Reviewer recommendation. The manuscript was carefully edited concerning English and the consistency of the text and figures. Conclusions have been significantly expanded. In our opinion  the list of references is adequate.

Reviewer 2 Report

Comments and Suggestions for Authors

Author Response

The paper reports on adaptation of standard sample holders (PTS and FLAG) to enable applying electrical and magnetic fields (as well as strains) to the sample hold in ultra-high vacuum. As for applications of the improved holders, the authors reproduce data from their earlier publications [20,21] together with previously unpublished results. My comments are: 1. How do you measure the magnitudes of the magnetic fields? What are uncertainties of the values (of some tens of mT) reported in the text? 2. The same question about the electric fields. Of course, you can easily measure the bias in volts, but did you estimate the influence of the sample and contact geometry? Or, maybe, you mean bias voltage dependence (of, say, deposition rate) rather than the dependence on electric field strength? 3. A similar question on controlling the strains. Do you measure the bending curvature or some other stress parameters? 4. Figure 15b: What is the uncertainty of the data points? Some minor issues: • Line 313: space after [21] • Line 332: Chinese inscription after [21] • Line 424: marks

ad. 1. Thank you very much for this remark; we overlooked this issue in the manuscript. The magnetic field induction and its distribution were carefully measured ex situ and, in the case of the variable field adapters,  calibrated against the coil current. A corresponding paragraph was added:

“The “Magnetic field range” data are for reference only. The exact magnetic field induction and its distribution were measured in the air for a particular adapter and, in the case of the variable field holders, calibrated against the coil current. For the permanent magnet adapters, the MF uniformity contributes the most to the characteristic uncertainties, which for the sample area of 5x5 mm2 did not exceed 5%. Additionally, the thickness of the inserted FLAG-substrate combination contributed to an additional minor MF uncertainty. The maximum magnetic field, its uniformity, and uncertainty for the electromagnet solution with in-plane MF are determined by the electromagnet gap. The values in Table 1 are for the 10 mm gap, ensuring 3%-field uniformity, sufficient for measuring wedge samples along the 5 mm distance, as it was verified by the comparison of MOKE measurements using the PTS adapter and a laboratory electromagnet. A reasonable reduction of the gap to 5 mm allows the MF induction to be doubled at the expense of uniformity.”

ad.2. Concerning the electric field holder, we give only the values of the maximum voltage (1000 V) limited by the cable insulation. The exact parameters of the electric field and its gradient in deposition under the influence of EF (which is planned) will be calculated for the given electrode geometry, as mentioned in the text.  Here, we report only transport measurements where the voltage matters.

 ad.3. Again, thank you for pointing out the omitted data, which have been now included:

“The maximum curvature is determined by the piston stroke and the clamp distance, which are 0.5 mm and 6 mm, respectively, which gives the minimum curvature radius as small as approximately 10 mm.”

ad. 4. The experimental error is less than the point size. The corresponding remark was added to the figure caption.

All minor issues were corrected.

Reviewer 3 Report

Comments and Suggestions for Authors

The manuscript outlines the novel integration of electric, magnetic, and strain fields in the MBE process to improve and control the growth of epitaxial layers and nanostructures. Specifically, it highlights three aspects: electric fields can enhance surface diffusion and reduce defects through electromigration; how magnetic fields influence the composition and magnetic properties of thin films; and impact of strain fields on the structural and magnetic properties due to lattice mismatches and thermal expansion differences. This approach demonstrates significant advancements in achieving optimized material properties by addressing limitations in traditional MBE techniques, which typically rely on substrate temperature, deposition rate, and gas partial pressures. However, the manuscript could benefit from a deeper mechanistic insight into how these external fields interact with the atomic processes during growth, a broader scope of materials studied, and a more comprehensive quantitative analysis to substantiate the benefits of this method.

The manuscript briefly mentions the effects of external fields but lacks detailed mechanistic explanations. Provide a theoretical framework that explains how electric fields enhance surface diffusion or how magnetic fields alter nucleation and growth processes. Specifically, expand on the discussion in Paragraphs 42-47 and Paragraphs 67-77, where the effects of electric and magnetic fields are initially mentioned. These paragraphs should include more detailed explanations of the underlying physical mechanisms at the atomic level.

Provide more quantitative data about the morphology properties of films grown with and without the magnetic fields the in the figure 11 STM measurement. For example, quantify the improvements in crystallinity, reduction in defects, surface smoothness, etc.

The manuscript focuses on a limited range of materials, primarily Fe3O4 and Co film in a Pt/Co/Pt/MgO(111) films. It would be worthy to expand the scope to include other materials and discuss potential applications in different fields, such as semiconductor devices, spintronic materials, and advanced sensors, demonstrating the versatility of the proposed method.

The conclusion paragraph is well too short. Please reaffirm the key points and arguments in this conclusion to deliver clear message to the reader. Also, it would be worthy to add future research directions and potential industrial applications in the conclusion section. For example, discuss how the technology can be scaled up for mass production, integrated into existing manufacturing processes, and explore the long-term stability and performance of the materials fabricated using this method.

Author Response

The manuscript outlines the novel integration of electric, magnetic, and strain fields in the MBE process to improve and control the growth of epitaxial layers and nanostructures. Specifically, it highlights three aspects: electric fields can enhance surface diffusion and reduce defects through electromigration; how magnetic fields influence the composition and magnetic properties of thin films; and impact of strain fields on the structural and magnetic properties due to lattice mismatches and thermal expansion differences. This approach demonstrates significant advancements in achieving optimized material properties by addressing limitations in traditional MBE techniques, which typically rely on substrate temperature, deposition rate, and gas partial pressures. However, the manuscript could benefit from a deeper mechanistic insight into how these external fields interact with the atomic processes during growth, a broader scope of materials studied, and a more comprehensive quantitative analysis to substantiate the benefits of this method.

The manuscript briefly mentions the effects of external fields but lacks detailed mechanistic explanations. Provide a theoretical framework that explains how electric fields enhance surface diffusion or how magnetic fields alter nucleation and growth processes. Specifically, expand on the discussion in Paragraphs 42-47 and Paragraphs 67-77, where the effects of electric and magnetic fields are initially mentioned. These paragraphs should include more detailed explanations of the underlying physical mechanisms at the atomic level.

Provide more quantitative data about the morphology properties of films grown with and without the magnetic fields the in the figure 11 STM measurement. For example, quantify the improvements in crystallinity, reduction in defects, surface smoothness, etc.

The manuscript focuses on a limited range of materials, primarily Fe3O4 and Co film in a Pt/Co/Pt/MgO(111) films. It would be worthy to expand the scope to include other materials and discuss potential applications in different fields, such as semiconductor devices, spintronic materials, and advanced sensors, demonstrating the versatility of the proposed method.

The conclusion paragraph is well too short. Please reaffirm the key points and arguments in this conclusion to deliver clear message to the reader. Also, it would be worthy to add future research directions and potential industrial applications in the conclusion section. For example, discuss how the technology can be scaled up for mass production, integrated into existing manufacturing processes, and explore the long-term stability and performance of the materials fabricated using this method.

The aim of our manuscript, as clearly represented by the title, was to introduce the novel method MBE assisted by external magnetic, electric, and strain fields and give examples proving that the technique works.  In our opinion, the goal of presenting the new technology was achieved. Deeper insight into the physics of the interaction of the external field with the film matter is out of the scope of the present manuscript.   We fully agree with the Reviewer that a broad class of materials may benefit from using the method, and a program of extended studies using the method is envisioned and partially in progress, as will be reported in future publications. Therefore, we refrain from complying with the reviewer's requests to expand the scope of the manuscript to other materials because it is simply impossible to do so within the limit of the present submission. On the other hand, we follow the request to expand the concluding paragraph in line with the reviewer’s recommendation, as cited below:

"The developed sample holders give numerous new functionalities of the MBE system, as shown in application examples related to the magnetic and electric properties of selected epitaxial systems. The external stimuli integrated into the adapters and sample holders allow the extension of the sample environment during the MBE process and expand the range of in situ experimental tools. Presented solutions can be easily implemented in existing UHV systems and allow ingenious combination of the PTS and FLAG holder standards. The developed research-laboratory-oriented solution has some direct application potential; however, adapting the present technology to industrial applications might be challenging due to the requirements imposed on the large size of the substrates.

We expect that the developed methodology will help precisely tailor the functionality of epitaxial nanostructures for a wide range of applications, particularly spintronics. Our research hypothesis is that activation of these external stimuli during MBE growth will affect many structural, electronic, and magnetic properties. The background of our hypothesis is a high susceptibility of the MBE process to external parameters on the one hand and strong dependence of the physical parameters on the system size and dimensionality, known as size effects, on the other hand. Thus, when an external stimulus is applied during the film growth, it is plausible that a critical temperature (e.g. the Curie or Neel temperature) for an ultrathin film will fall in the deposition temperature range, resulting in an enhanced impact of the applied field. The row of materials in the scope of our interest and being under current investigations includes iron oxides (hematite, magnetite), metal-oxide heterostructures including a ferromagnetic layer with perpendicular magnetic anisotropy with an antiferromagnetic layer responsible for the exchange bias effect. The researched problems in progress are, for example, tuning the structural phase transition in hematite by MF-assisted growth, controlling the growth of epitaxial oxide films on flexible substrates by EF and SF, tailoring of perpendicular magnetic anisotropy in epitaxial heterostructures by MF-assisted MBE. Flexible oxide films (free-standing or deposited on flexible substrates) are an emerging promising technology in electronics. Spinel films (magnetite or cobalt ferrite) on mica belong to the most studied epitaxial systems [24]; however, detailed analysis of the role of strain/stress on growth, magnetoelastic, and transport properties is lacking. Using in situ bending technics, we will be able to quantify these effects and contribute to the fundamental understanding of the mechanism behind them."